# DefNTaxS: The Inevitable Need for Taxonomic Definition in Classification

## Abstract

Existing approaches leveraging large pretrained vision-language models (VLMs) like CLIP for zero-shot text-image classification often focus on generating fine-grained class-specific descriptors, leaving higher-order semantic relations between classes underutilized. We address this gap by proposing **Defi**Ned **Tax**onomic **S**tratification (**DefNTaxS**), a novel and malleable framework that supplements per-class descriptors with inter-class taxonomies to enrich semantic resolution in zero-shot classification tasks. Using large language models (LLMs), DefN-TaxS automatically generates subcategories that group similar classes and appends context-specific prompt elements for each dataset/subcategory, reducing inter-class competition and providing deeper semantic insight. This process is fully automated, requiring no manual modifications or further training for any of the models involved. We demonstrate that DefNTaxS yields consistent performance gains across a number of datasets often used to benchmark these frameworks, enhancing accuracy and semantic interpretability in zero-shot classification tasks of varying scale, granularity, and type.

## 1 Introduction

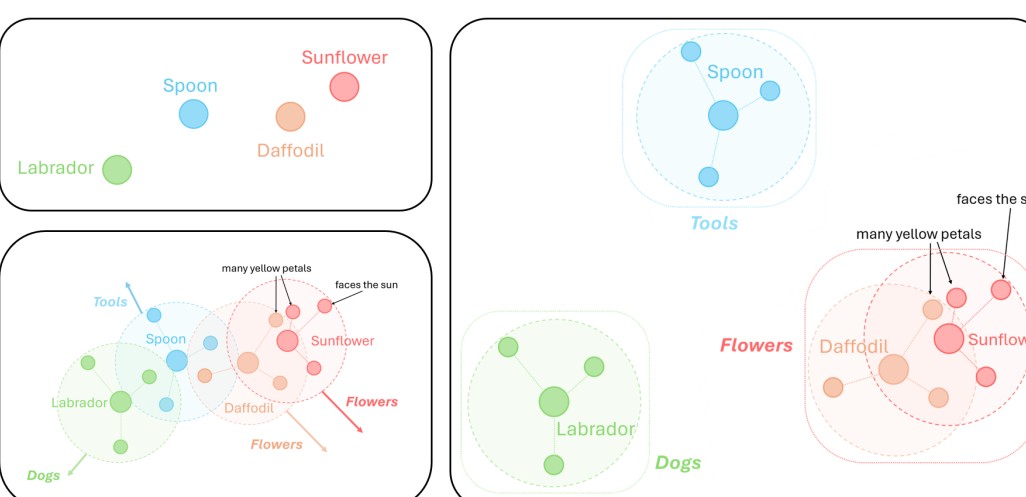

Figure 1: Conceptual visualization of the difference in embedding geometries using CLIP, D-CLIP, and DefNTaxS. While CLIP relies on class names for classification, D-CLIP uses class-specific descriptors to enhance classification accuracy. DefNTaxS further improves classification by incorporating taxonomic subcategories to reduce inter-class competition and enhance semantic resolution. The structured taxonomic information provided by DefNTaxS helps differentiate classes at multiple levels of granularity, leading to more accurate and interpretable classification.

The rise of Vision-Language Models (VLMs) like CLIP (Radford et al., 2021) has transformed zero-shot text-image classification by learning shared representations between visual content and textual descriptions. These models effectively align multimodal data, enabling quick classification of

images based on text prompts without additional training. However, their performance heavily relies on the specificity of these prompts, often making it challenging to distinguish between semantically similar classes.

To address this, recent approaches like (Menon & Vondrick, 2023; Pratt et al., 2023; Novack et al., 2023) have used Large Language Models (LLMs) to generate detailed class-specific descriptors, enhancing text-image alignment. WaffleCLIP Roth et al. (2023) achieves similar accuracy to D-CLIP by replacing LLM-generated descriptors with random words, highlighting that high-level semantic concepts from LLMs enhance classification more effectively than fine-grained details. CuPL (Pratt et al., 2023) also uses LLMs for generating descriptors, but while D-CLIP enforces a structured list of identifying features to improve explainability, it may reduce classification performance; CuPL, in contrast, employs multiple free-form prompts to capture nuanced category information, resulting in improved accuracy. CHiLS (Novack et al., 2023) takes a different approach by refining class labels into finer-grained subclasses, using either existing label hierarchies or LLMs like GPT-3 to generate linguistic hyponyms, whereas our work clusters related classes into broader taxonomic groups to streamline classification and reduce competition among similar classes. MPVR (Mirza et al., 2024) leverages LLMs to automate the creation of diverse, category-specific prompts for zero-shot image recognition based on minimal input such as task descriptions and class labels. While effective to some extent, these methods face limitations: (1) They overly focus on fine-grained details, neglecting medium- and coarse-grained semantics that provide crucial context. (2) Fine-grained descriptors can introduce noise and ambiguity, reducing interpretability and leading to misclassifications. (3) A lack of structured semantic hierarchy amplifies competition between similar classes, particularly in datasets with high intra-class similarity.

Motivated by viewing zero-shot classification through the lens of competition among classes, we argue that the goal is not to find the "best" descriptor for a class, but rather the "most distinctive" one. This perspective aligns with the idea that classes should not directly compete with one another in a complex, high-dimensional space. Instead, effective differentiation can be achieved by grouping classes within a structured hierarchy, leveraging taxonomic relationships to enhance clarity. By working together within this framework and establishing distinctions at multiple levels of resolution, classes can reduce inter-class competition and improve classification accuracy.

To this end, we propose Defined Taxonomic Stratification (DefNTaxS), a novel approach designed to overcome the limitations of existing methods by incorporating taxonomy classes directly into the text prompts of CLIP. Our method uses LLMs to analyze the classes within a dataset and propose taxonomic groupings based on shared semantic relationships, creating a hierarchical classification framework. DefNTaxS consistently outperforms existing methods across all evaluated datasets, showcasing its effectiveness in zero-shot classification tasks. Additionally, it reveals and organizes the underlying structure of the CLIP embedding space, offering a semantically structured view that clarifies how classes are organized and differentiated within the hierarchy.

## 2 RELATED WORK

**Zero-shot Image Classification using VLMs.** Vision-Language Models (VLMs) (Jia et al., 2021; Kim et al., 2021; Radford et al., 2021; Yao et al., 2022; Wang et al., 2022; Yu et al., 2022; Cho et al., 2021; Li et al., 2023; Naeem et al., 2023) learn a joint representation that aligns visual content with associated textual descriptions in a shared embedding space. This learned alignment allows VLMs to perform effectively on zero-shot image classification tasks, where they rely on textual cues, such as class labels, to classify novel image categories without prior exposure during training. Notably, CLIP (Radford et al., 2021) has emerged as a prominent approach for learning multimodal representations that align visual and textual information within a shared embedding space. The model utilizes a dual-encoder architecture, with separate encoders for image and text modalities, trained through contrastive learning to maximize the similarity between matching image-text pairs and minimize it for non-matching pairs. Each encoder can have a different backbone. At inference, CLIP uses prompts like "a photo of a `[class name]`" providing context for classification and enabling zero-shot transfer to various tasks without task-specific fine-tuning. Subsequent works, such as FLAVA (Singh et al., 2022), Florence (Yuan et al., 2021), and BLIP (Li et al., 2022), have built upon the CLIP paradigm and advanced multimodal representation learning. Florence enhances this learning by leveraging a significantly larger and more diverse pre-training dataset. FLAVA focuses

on novel training objectives beyond contrastive learning, such as masked image modeling combined with contrastive loss, to improve multimodal understanding. Meanwhile, BLIP incorporates a refined model architecture that better integrates visual and linguistic features for more effective joint representation. VLM research follows two main pipelines: visual prompting and text prompting. Visual prompting enhances performance by processing or aligning visual inputs with textual representations, while text prompting focuses on refining textual descriptors Li et al. (2024); Zhang et al. (2024). Our work adopts an exclusively text-based approach, leaving the images, model weights, and embeddings unaltered.

**Training-free textual prompting in VLMs.** While CLIP demonstrates strong zero-shot capabilities, its performance in downstream tasks is significantly affected by prompt choice, as noted by (Radford et al., 2021) and (Zhou et al., 2022). (Zhou et al., 2022) specifically point out that finding the optimal prompt is a complex and time-consuming process, often requiring prompt tuning. However, with the rise of large language models (LLMs) like GPT-3 (Brown, 2020), new approaches (Menon & Vondrick, 2023; Pratt et al., 2023) have emerged to enhance CLIP's zero-shot generalization by leveraging LLMs. Rather than relying on handcrafted templates to generate class features, these methods utilize LLMs to create high-level concepts, class descriptions resulting in enriched text features and improved performance. D-CLIP (Menon & Vondrick, 2023) demonstrated that leveraging the knowledge embedded in LLMs to automatically generate class-specific descriptions that focus on the discriminating features of image categories can enhance zero-shot classification. WaffleCLIP (Roth et al., 2023) achieves the same accuracy as D-CLIP by replacing LLM-generated descriptors with random words. It highlights that high-level semantic concepts from LLMs improve classification more effectively than fine-grained details. CuPL (Pratt et al., 2023) also uses LLMs for generating descriptors, but D-CLIP enforces a structured list of identifying features, enhancing explainability but potentially reducing classification performance. In contrast, CuPL generates multiple, free-form prompts to better capture the nuances of each category, resulting in improved accuracy. CHiLS (Novack et al., 2023) refines class labels into finer-grained subclasses by leveraging either existing label hierarchies or large language models like GPT-3 to generate linguistic hyponyms for each class. In our work, we also consider the taxonomy of classes but take the opposite approach—by clustering related classes into broader taxonomic groups to reduce competition among similar classes and streamline classification. MPVR (Mirza et al., 2024) automates the creation of category-specific prompts for zero-shot image recognition by leveraging LLMs to generate diverse prompts based on minimal input, such as a task description and class labels. Another study Ren et al. (2024) addresses zero-shot classification by constructing a class hierarchy through iterative k-means clustering and LLM-generated descriptions; in contrast, our work avoids clustering and iterative refinement, offering a more efficient, single-stage framework with enhanced semantic interpretability through directly leveraging inter-class taxonomies.

## 3 METHOD

### 3.1 GENERATING SUBCATEGORIES

The aim of this approach is to enhance zero-shot classification performance by reducing unnecessary competition amongst classes. This problem arises when each class is considered in direct competition with all other classes, which can result in misclassification, particularly for classes with overlapping semantics. To address this, our process begins by using the LLM to analyze the classes in the dataset and propose a set of taxonomic classes. These taxonomic classes are designed to cluster classes based on shared semantic relationships, thereby minimizing the competition between classes with similar characteristics. For instance, classes such as "forks," "knives," and "spoons" might all be grouped under a broader taxonomic class like "kitchen utensils."

Let $\mathcal{C} = \{c_1, c_2, \ldots, c_m\}$ be the set of classes in the dataset, with $m$ being the total number of classes. The LLM generates a set of taxonomic classes $T_c = \{t_1, t_2, \ldots, t_k\}$, where each $t_i \subseteq \mathcal{C}$ represents a subcategory grouping semantically related classes, as in Figure 1. Formally,

$$T_c = \{t_i \mid t_i \subseteq \mathcal{C}, \forall i = 1, \ldots, k\}, \tag{1}$$

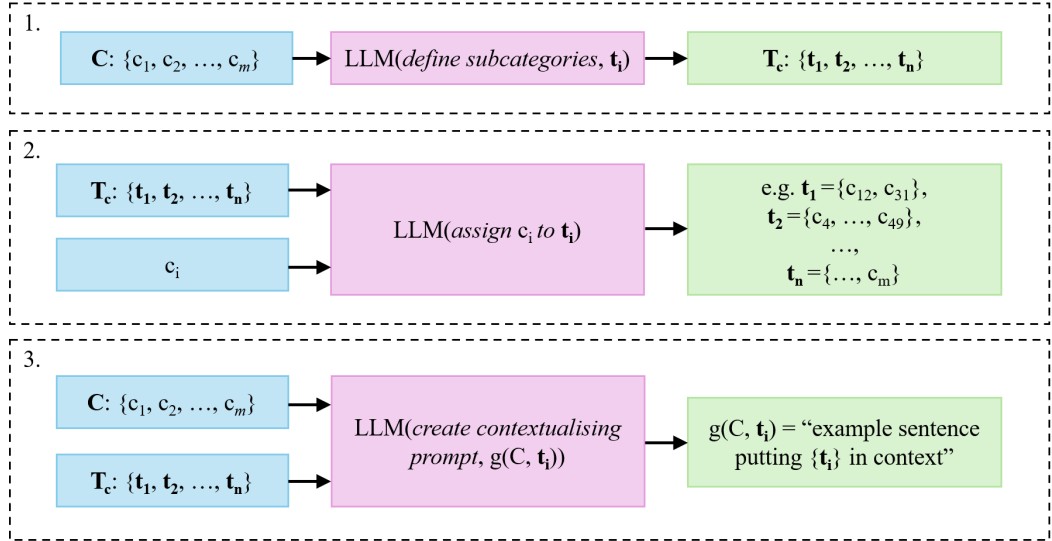

Figure 2: The creation of subcategories, the assignment of classes to them, and the generation of taxonomic class contextualizing sentences is completed iteratively using the LLM. Inputs are in blue, processes are in purple, and outputs are in green. (1) The LLM generates a set of taxonomic classes based on the classes in the dataset. If too few taxonomic classes are generated (i.e. $|T| < |C|/10$), the process is repeated. (2) Each class is assigned to one of the taxonomic classes. If too many classes are assigned to a single taxonomic class (i.e. $|t_i| > 20$), the process is repeated. (3) A sentence contextualizing the taxonomic class within the final prompt structure is generated.

where each $t_i$ is a subset of $\mathcal{C}$ such that

$$\bigcup_{i=1}^{k} t_i = \mathcal{C} \quad \text{and} \quad t_i \cap t_j = \emptyset \quad \forall i \neq j. \tag{2}$$

This means that the LLM creates a structured grouping where each taxonomic class is a non-overlapping subcategory of the original classes, covering all classes without redundancy. As in 2, each query to the LLM focuses exclusively on a single task to avoid confusion of the request or the model missing elements of the request. The prompts used also emphasize this necessity, as shown in Appendix A. This ensures that all classes will be assigned to a subcategory and only one subcategory.

If the size of the set of taxonomic classes $|T_c|$ is less than $\frac{|\mathcal{C}|}{10}$, then the list of taxonomic classes is provided back to the LLM for further refinement. Formally, if:

$$|T_c| < \frac{|\mathcal{C}|}{10} \tag{3}$$

then a refined set of taxonomic classes $T'_c = \{t'_1, t'_2, ..., t'_{k'}\}$ is generated, where $k' > k$. For instance, a taxonomic class like "dogs" may be further divided into more specific subcategories, such as "small dogs," "medium dogs," and "big dogs." This choice is validated empirically through repeatedly reducing the minimum number of subcategories to be generated, offering the LLM a greater number of options for assigning the classes. An example of this validation can be seen in Table 3 in 6.1.

Additionally, if the total number of classes is less than 20, $|\mathcal{C}| < 20$, we also consider a scenario where taxonomic class is assigned to be the name of the dataset. That is, the set of taxonomic classes is replaced by the name of the dataset:

$$T_c = \mathcal{C}. \tag{4}$$

For example, in the EuroSAT (Helber et al., 2017) dataset, instead of generating new subcategories, the taxonomic class contextualizing sentence ending is replaced with "from the EuroSAT dataset".

## 3.2 DESCRIPTORS WITH SEMANTIC CONTEXT

Once the list of potential taxonomic classes $T_c$ is finalized, we iteratively use the LLM to allocate each individual class $c \in \mathcal{C}$ to one of the taxonomic classes $t_i \in T_c$. This allocation provides a semantic context for each class based on its shared relationships with other classes within the subcategory. Consequently, each class $c$ is not only associated with its specific textual descriptors $D_c = \{d_1, d_2, ..., d_{|D_c|}\}$, but also with the broader context of its taxonomic class $t$.

To effectively encode this semantic information into a text prompt, we construct a structured text representation for each class $c$. Inspired by CLIP (Radford et al., 2021), which classifies a query image $x$ by finding the category $c \in \mathcal{C}$ that maximizes the cosine similarity between its image embedding $\phi_I(x)$ and its textual prompt embedding $\phi_L(f(c))$, where $f(c) = $ "A photo of a $\{c\}$", our approach enhances this structure. Specifically, D-CLIP (Menon & Vondrick, 2023) introduces a richer set of descriptors $D_c$, using prompts of the form $f(c,d) = $ "$\{c\}$ which is/has/etc $\{d\}$" to better capture the visual characteristics of each category. The classification score of D-CLIP is computed by averaging the similarity between the image embedding and all descriptor embeddings:

$$\tilde{c} = \arg\max_{c \in \mathcal{C}} \frac{1}{|D_c|} \sum_{d \in D_c} s(\phi_I(x), \phi_L(f(c,d))), \tag{5}$$

where $s(\cdot, \cdot)$ denotes the cosine similarity.

DefNTaxS further modifies this by incorporating the taxonomic class context $T_c$. For each class $c$, the corresponding taxonomic class $t_i \in T_c$ is included in the textual prompt to provide a broader semantic context. Specifically, we introduce a function $g(\mathcal{C}, t_i)$ that generates a sentence contextualizing the taxonomic class within the dataset. For example, the Food101 dataset may have the contextualizing sentence, $g(\mathcal{C}, t_i) = $ "on a menu under "$t_i$"", and the class "cannoli" with descriptor "nuts" may be assigned to the subcategory "desserts", producing:

$$f(c, d, g(\mathcal{C}, t_i)) = \text{"cannoli, which has nuts, found on a menu under "desserts"".} \tag{6}$$

The classification score is then calculated by averaging the similarity between the image embedding and all descriptor embeddings that incorporate the taxonomic context:

$$\tilde{c} = \arg\max_{c \in \mathcal{C}} \frac{1}{|D_c|} \sum_{d \in D_c} s(\phi_I(x), \phi_L(f(c, d, g(\mathcal{C}, t_i)))). \tag{7}$$

By introducing $g(\mathcal{C}, t_i)$, the textual prompt effectively leverages both the class-specific descriptors and the broader semantic relationships defined by the taxonomic classes, improving the model's ability to capture complex inter-class relationships in zero-shot classification.

## 4 EXPERIMENTAL SETTINGS

In this section, we assess the performance of the DefNTaxS method through a series of experiments and comprehensive ablation studies.

### 4.1 IMPLEMENTATION/EVALUATION DETAILS

Unless specified otherwise, all experiments are conducted on a single NVIDIA RTX 4090 GPU. The descriptors used in the experiments are sourced from the prior work in D-CLIP (Menon & Vondrick, 2023). The prompt structure used in the experiments is the same as that of D-CLIP (Menon & Vondrick, 2023), which follows the format "$c_i$ which has/is $d_i$." The models are evaluated using the same zero-shot classification setup as in (Menon & Vondrick, 2023), with the same train-test splits and evaluation metrics. The classification accuracy is reported as the primary evaluation metric, with additional analysis provided to understand the impact of the proposed method on the model's decision-making process.

### 4.2 DATASETS

For evaluating our method, we use the benchmark outlines provided in Menon & Vondrick (2023) for zero-shot classification. This benchmark consists of ImageNet (Deng et al., 2009), a dataset

for classifying everyday objects; CUB (Welinder et al., 2010), which focuses on fine-grained bird species classification; Oxford Pets (Parkhi et al., 2012), designed for the recognition of common pets; DTD (Cimpoi et al., 2014), used for texture and pattern classification in natural settings; Food101 (Bossard et al., 2014), aimed at food categorization; and Places365 (Zhou et al., 2017), a large-scale dataset for scene and environment recognition. Furthermore, we assess our method on additional datasets such as EuroSAT (Helber et al., 2017), which focuses on land use and land cover classification based on Sentinel-2 satellite imagery.

## 4.3 BASELINES

In these experiments, we compare the performance of DefNTaxS against several state-of-the-art methods for zero-shot image classification using VLMs. The baselines include:

- **CLIP** (Radford et al., 2021), which uses the format "{class}" as the prompt,
- **E-CLIP** (Radford et al., 2021), an approach that enhances CLIP by using handcrafted templates for each class, such as "A photo of a {class}",
- **D-CLIP** Menon & Vondrick (2023), which generates class-specific descriptors using LLMs and uses the prompt format "{class} which has/is {descriptor}",
- **WaffleCLIP** (Roth et al., 2023), which replaces the LLM-generated descriptors with random words, using the format "{class} which has/is {random words/characters}",
- **WaffleCLIP + Concepts** (Roth et al., 2023), which uses the same structure as WaffleCLIP but includes high-level semantic concepts from LLMs, and
- **CuPL** (Pratt et al., 2023), which generates multiple free-form prompts for each class to capture the nuances of each category, with no specific format.

Each of these baselines was recreated using the setup described in 4.1 and the code provided for each study. All potential variables were maintained strictly to those used in the original studies. In doing so, we aimed to reduce any inconsistencies due to hardware, software, or other issues.

## 5 RESULTS

### 5.1 ZERO-SHOT CLASSIFICATION RESULTS

Table 1: Comparison of zero-shot visual classification performance across different image classification benchmarks using multiple CLIP backbones.

| Method | ImageNet | | | CUB | | | Oxford Pets | | | DTD | | | Food101 | | | Places365 | | | EuroSAT | | |
|---|---|---|---|---|---|---|---|---|---|---|---|---|---|---|---|---|---|---|---|---|---|
| | B/32 | B/16 | L/14 | B/32 | B/16 | L/14 | B/32 | B/16 | L/14 | B/32 | B/16 | L/14 | B/32 | B/16 | L/14 | B/32 | B/16 | L/14 | B/32 | B/16 | L/14 |
| CLIP | 58.89 | 64.10 | 71.55 | 51.86 | 56.42 | 62.98 | 77.88 | 80.14 | 86.82 | 41.12 | 44.57 | 50.74 | 77.83 | 84.02 | 89.87 | 37.50 | 38.32 | 39.04 | 44.26 | 46.10 | 36.83 |
| E-CLIP | 61.90 | 66.60 | 72.81 | 52.00 | 55.89 | 62.65 | 82.10 | 85.51 | 91.81 | 43.07 | 43.62 | 51.42 | 78.78 | 84.88 | 89.78 | 39.13 | 39.19 | 39.76 | 33.44 | 52.74 | 54.04 |
| CuPL | 62.12 | 66.01 | 73.68 | 52.34 | 56.84 | 63.03 | 81.78 | 84.03 | 84.60 | 90.95 | 42.61 | 43.87 | 79.84 | 83.89 | 88.97 | 38.87 | 39.01 | 38.57 | 41.50 | 38.57 | 48.25 |
| D-CLIP | 63.00 | 68.05 | 75.00 | 53.21 | 57.49 | 64.52 | 81.84 | 85.58 | 91.15 | 43.62 | 45.51 | 54.59 | 80.43 | 85.55 | 90.33 | 39.84 | 40.55 | 40.86 | 47.36 | 51.95 | 49.98 |
| WaffleCLIP | 62.35 | 67.29 | 74.07 | 52.17 | 56.20 | 62.34 | 82.38 | 81.22 | 88.24 | 40.05 | 42.50 | 49.41 | 79.43 | 85.27 | 90.51 | 38.35 | 39.52 | 39.86 | 31.49 | 31.94 | 34.28 |
| WaffleCLIP+concepts | 62.35 | 67.29 | 74.07 | 52.47 | 56.90 | 62.55 | 85.40 | 86.93 | 92.76 | 40.05 | 42.50 | 49.41 | 81.25 | 86.10 | 90.87 | 40.22 | 40.52 | 41.02 | 40.81 | 41.27 | 50.01 |
| DefNTaxS | 63.48 | 68.03 | 75.03 | 54.00 | 58.15 | 63.93 | 86.09 | 89.31 | 93.71 | 45.89 | 47.38 | 52.75 | 81.26 | 86.40 | 90.93 | 40.00 | 41.09 | 41.81 | 57.22 | 56.51 | 59.68 |

In this section, we present the zero-shot classification results of the DefNTaxS method compared to the baseline approaches on various benchmark datasets. The results are summarized in Table 1.

We observe that DefNTaxS approximately equals or outperforms the baseline methods across all datasets, achieving higher classification accuracy with a method that requires no additional training or manual intervention. The improvements are particularly pronounced on datasets with high class counts or high intra-class similarity, where the taxonomic grouping helps reduce inter-class competition and improve classification accuracy.

Due to inherent ambiguity in many class labels, many approaches of this type (Menon & Vondrick, 2023; Roth et al., 2023) require extra context be manually added to more accurately capture the expected content of the images within these datasets. For example, a dog and a fighting athlete may both be described as "a boxer", but may suffer from classification deterioration unless specified. DefNTaxS naturally solves many of these issues through the generated subcategory titles and

their common ability to capture this specificity. This is a significant improvement in the processing required to achieve the results, completely eliminating

One exception to this subcategory contextualization is with the EuroSAT dataset Helber et al. (2017), where the small number of classes leads us to automatically default to using the dataset name to contextualize the prompt. For completeness, the contextualizing sentence shown in 3.1 was replaced by "from a dataset of satellite images." and also achieved the significant result of 55.13% accuracy with ViT-B/32.

In other cases, we see benefits due to common co-appearing text structures(Udandarao et al.). As an example, images of pets tend to be uploaded more often in a casual, social location, often appearing with simple statements like "this is a photo of [pet's name]". For this reason and with no other benchmark, we see an improved performance with the Oxford Pets dataset (Parkhi et al., 2012) when prefixing the classification prompts with the standard CLIP templates, which often capture these simple statements that often appear on Facebook statuses, Instagram captions, and other popular image sharing sites.

## 5.2 DOMAIN GENERALIZATION RESULTS

To understand the impact of the proposed method on out of domain generalization, we evaluate the performance of DefNTaxS on the ImageNetV2 dataset, which is designed to test the generalization capabilities of models trained on ImageNet. The dataset matches the distribution frequency of the original ImageNet dataset but contains new images, making it a suitable benchmark for assessing the model's ability to generalize to unseen data. We compare the performance of DefNTaxS against this baseline to assess the model's generalization capabilities in Table 2.

Table 2: Comparison of zero-shot visual classification performance on the ImageNetV2 dataset using three different CLIP backbones (B/32, B/16, L/14).

| Method | ImageNet V2 | | |
| --- | --- | --- | --- |
| | B/32 | B/16 | L/14 |
| CLIP | 51.70 | 57.86 | 65.43 |
| E-CLIP | 54.45 | 60.62 | 67.14 |
| D-CLIP | 55.77 | **61.54** | **69.33** |
| WaffleCLIP | 52.98 | 58.64 | 65.67 |
| DefNTaxS | **56.31** | 61.49 | 68.84 |

We observe that DefNTaxS outperforms the baseline for this task to a similar scale as the original ImageNet dataset, demonstrating the effectiveness of the proposed method in improving the model's generalization capabilities.

## 6 ABLATION

Factors that were considered in the ablation study include the structure of the prompt, the length of the prompt, the number of subcategories generated, and the impact of the taxonomic class on classification performance. This study aims to provide insights into the effectiveness of the proposed method and identify the key components that contribute to its performance.

### 6.1 REDUCED TAXONOMIC REFINEMENT

For larger datasets, especially ImageNet and Places365 with hundreds of classes, the taxonomic refinement process may result in subcategories with a large number of classes. For example, a subcategory like "dogs" could contain over 150 different dog species in ImageNet, leading to increased competition between classes within the same subcategory. To investigate the impact of this scenario, we conduct an ablation study where the taxonomic refinement process is limited to a single iteration, resulting in subcategories with 100 or more classes. We also conducted various studies of gradually increasing the number of taxonomic subcategories, but as the numbers purely act as

a guide for the LLM in generating these subcategory names (Appendix A), the results varied insignificantly from the results of this main study. The results of this ablation study, shown in Table 3, demonstrate a significant decrease in classification accuracy, indicating that the model gains little benefit from subcategories that are coincident with a large number of classes, as it can provide distinction between the classes. This highlights the importance of refining the taxonomic structure to create more distinct subcategories, which can help reduce inter-class competition and improve classification performance.

Table 3: Effect of reduced taxonomic refinement on zero-shot visual classification performance for the ImageNet and Places365 datasets.

| Method | ImageNet | | | Places365 | | |
|---|---|---|---|---|---|---|
| | B/32 | B/16 | L/14 | B/32 | B/16 | L/14 |
| CLIP | 58.86 | 64.07 | 71.57 | 37.48 | 38.33 | 39.05 |
| E-CLIP | 61.90 | 66.61 | 72.80 | 39.12 | 39.18 | 39.75 |
| D-CLIP | **63.26** | **68.38** | **75.16** | **40.89** | **41.85** | **41.46** |
| WaffleCLIP | 60.25 | 64.60 | 71.91 | 38.28 | 38.05 | 38.93 |
| DefNTaxS | 61.23 | 66.14 | 74.72 | 37.53 | 40.22 | 39.89 |

## 6.2 DESCRIPTOR REGENERATION

In this ablation study, we investigate the impact of regenerating the descriptors using more advanced LLMs, such as GPT-4 or other state-of-the-art models, to determine if this process provides similar benefits to the taxonomic refinement. We intend to understand whether the improvements in classification performance are primarily due to the subcategories or the enhanced semantic information within descriptors generated by more powerful LLMs.

The results show in Table 4 that regenerating the descriptors with more advanced LLMs does not provide the same benefits as the taxonomic refinement process. However, the combination of both approaches leads to a significant improvement in classification accuracy, suggesting that the subcategories and enhanced descriptors complement each other to enhance the model's performance.

Table 4: Comparison of zero-shot visual classification performance across different image classification benchmarks using multiple CLIP backbones and descriptors generated by GPT-4.

| Method | ImageNet | | | CUB | | | Oxford Pets | | | DTD | | | Food101 | | | Places365 | | | EuroSAT | | |
|---|---|---|---|---|---|---|---|---|---|---|---|---|---|---|---|---|---|---|---|---|---|
| | B/32 | B/16 | L/14 | B/32 | B/16 | L/14 | B/32 | B/16 | L/14 | B/32 | B/16 | L/14 | B/32 | B/16 | L/14 | B/32 | B/16 | L/14 | B/32 | B/16 | L/14 |
| CLIP | 58.86 | 64.07 | 71.57 | 51.83 | 56.35 | 62.98 | 77.96 | 80.12 | 86.83 | 41.08 | 44.59 | 50.76 | 77.84 | 84.02 | 89.86 | 37.48 | 38.33 | 39.05 | 44.32 | 46.20 | 36.97 |
| E-CLIP | 61.90 | 66.61 | 72.80 | 51.95 | 55.87 | 62.70 | 82.06 | 85.49 | 91.87 | 43.12 | 43.60 | 51.44 | 78.79 | 84.86 | 89.78 | 39.12 | 39.18 | 39.75 | 33.31 | 52.56 | 54.10 |
| CuPL | 62.10 | 67.23 | 73.31 | 51.97 | 56.89 | 63.45 | 80.89 | 82.25 | 90.78 | 45.51 | 45.95 | 53.61 | 79.26 | 83.71 | 90.02 | 39.84 | 40.55 | 40.86 | 47.36 | 51.95 | 49.98 |
| D-CLIP | 63.26 | 68.38 | 75.16 | 53.83 | 59.13 | 65.34 | 81.54 | 85.64 | 91.58 | 47.11 | 47.64 | 56.54 | 81.06 | 86.09 | 91.22 | 40.89 | 41.85 | 41.46 | 42.80 | 49.85 | 46.08 |
| WaffleCLIP | 62.26 | 67.18 | 74.11 | 52.05 | 55.42 | 62.75 | 80.12 | 81.27 | 88.17 | 41.03 | 44.49 | 50.46 | 80.31 | 85.23 | 90.60 | 38.64 | 39.64 | 40.10 | 35.28 | 49.47 | 46.24 |
| WaffleCLIP+concepts | 62.26 | 67.18 | 74.11 | 52.49 | 56.18 | 63.13 | 85.33 | 86.64 | 93.88 | 41.03 | 44.49 | 50.46 | 81.56 | 86.41 | 91.28 | 40.62 | 40.82 | 41.25 | 46.05 | 49.39 | 51.59 |
| DefNTaxS | 63.63 | 68.28 | 75.05 | 54.42 | 59.53 | 64.62 | 86.67 | 89.27 | 93.14 | 48.09 | 48.87 | 54.57 | 81.47 | 86.45 | 91.28 | 39.31 | 40.81 | 40.96 | 57.51 | 60.25 | 60.67 |

## 6.3 DEFNTAXS WITHOUT DESCRIPTORS

Much research on hierarchical approaches to zero-shot text-image classification focus of either ascending or descending levels of descriptive resolution, but rarely both. DefNTaxS leverages both the benefits of greater taxonomic hierarchy while also incorporating the fine-grained visual descriptors introduced by D-CLIP (Menon & Vondrick, 2023). Comparison between the DefNTaxS approach and D-CLIP through the baselines in 1, isolating the effect of fine-grained semantic information, but for completeness we must also investigate the effect of the taxonomic subcategories.

In all but a select few cases, the original DefNTaxS approach outperforms both approaches that isolate a single factor: either fine-grained semantics or taxonomic hierarchy. However, the isolated taxonomies do show benefit over the CLIP baseline and even outperform all other baselines with the Food101 dataset Bossard et al. (2014). Results of this investigation can found in Table 5.

Table 5: Comparison of zero-shot visual classification performance between DefNTaxS and DefN-TaxS without the use of D-CLIP-based descriptors.

| Method | ImageNet | | | CUB | | | Oxford Pets | | | DTD | | | Food101 | | | Places365 | | | EuroSAT | | |
|---|---|---|---|---|---|---|---|---|---|---|---|---|---|---|---|---|---|---|---|---|---|
| | B/32 | B/16 | L/14 | B/32 | B/16 | L/14 | B/32 | B/16 | L/14 | B/32 | B/16 | L/14 | B/32 | B/16 | L/14 | B/32 | B/16 | L/14 | B/32 | B/16 | L/14 |
| CLIP | 58.89 | 64.10 | 71.55 | 51.86 | 56.42 | 62.98 | 77.88 | 80.14 | 86.82 | 41.12 | 44.57 | 50.74 | 77.83 | 84.02 | 89.87 | 37.50 | 38.32 | 39.04 | 44.26 | 46.10 | 36.83 |
| D-CLIP | 63.00 | 68.05 | 75.00 | 53.21 | 57.49 | 64.52 | 81.84 | 85.58 | 91.15 | 43.62 | 45.51 | 54.59 | 80.43 | 85.55 | 90.33 | 39.84 | 40.55 | 40.86 | 47.36 | 51.95 | 49.98 |
| DefNTaxS | 63.48 | 68.03 | 75.03 | 54.00 | 58.15 | 63.93 | 86.09 | 89.31 | 92.76 | 45.89 | 47.38 | 52.75 | 81.26 | 86.10 | 90.93 | 40.00 | 41.09 | 41.81 | 57.22 | 56.51 | 59.68 |
| DefNTaxS$_{ans_d}$escriptor | 62.30 | 66.09 | 73.34 | 53.94 | 57.42 | 62.75 | 85.25 | 88.78 | 92.75 | 43.49 | 44.17 | 49.61 | 81.37 | 86.36 | 90.60 | 38.87 | 39.40 | 40.04 | 55.17 | 51.97 | 45.34 |

## 6.4 PROMPT MODIFICATION

In this section, we present an ablation study aimed at systematically analyzing how the structure/-format and length of the prompt in the language component of CLIP (Radford et al., 2021) impact classification performance. To conduct this analysis, we utilize the CUB (Welinder et al., 2010) dataset, a fine-grained image dataset that provides a suitable context for evaluating the sensitivity of CLIP to variations in prompt design.

### 6.4.1 STRUCTURE OF PROMPT

Table 6: Impact of Different Prompt Structures on Zero-Shot Classification Accuracy

| Method | Prompt Structure | Accuracy (%) |
|---|---|---|
| E-CLIP Baseline | `"A photo of a {c}"` | 51.95 |
| D-CLIP Baseline | `"{c}, which is/has/etc {d}"` | 52.57 |
| Class-Descriptor Switch | `"{d}, which is/has/etc {c}"` | 51.34 |
| Prefix Modification | `"An image of a {c}, which has/is {d}"` | 50.94 |
| Class-Specific Prefix Modification | `"A photo of a {c}, which has/is {d}, a type of bird"` | 53.33 |
| Class Label Modification | `"{c}, which is/has/etc {d}"` | 22.14 |
| Descriptor-Only | `"{d}"` | 3.81 |
| Class Repetition | `"{c}, which is/has/etc {c}"` | 52.35 |

In our initial investigation, we analyzed the prompt structure of D-CLIP (Menon & Vondrick, 2023), which follows the format *"$c_i$ which has/is $d_i$,"* where $c_i$ represents the class and $d_i$ the descriptor. We explored how the arrangement of these elements influences classification performance.

We first tested reversing the positions of the class and descriptor, using the structure *"$d_i$, which is a description of a $c_i$."* This modification aimed to prioritize the descriptors over the class, based on findings that initial tokens in a prompt have greater weight in embedding space (Han et al., 2024; Kazemnejad et al., 2024). However, this change resulted in reduced accuracy, showing that the model performs better when the class is positioned at the start of the prompt.

Next, we added prefixes such as *"An image of ..."* before the class label, restructuring the prompt as *"An image of a $c_i$, which has/is $d_i$."* This modification also decreased accuracy, as the filler content shifted focus away from the class. The model consistently performed better when the class was positioned at the start of the prompt without additional prefixes. However, one notable exception was observed with domain-specific templates, such as the BirdSnap template. Using a structure like *"a photo of a class label, a type of bird,"* tailored for bird classification, significantly improved accuracy, even surpassing the baseline D-CLIP performance. This indicates that carefully designed, domain-specific templates can be beneficial despite generally negative effects of filler content.

We also experimented with simplifying class names to focus on broad categories. For instance, *"Red-winged Blackbird"* was simplified to *"Blackbird,"* relying on the descriptor to distinguish between similar classes. This approach significantly reduced accuracy, as it removed distinctive features from the class name and increased dependence on the descriptors, which were often insufficiently detailed for fine-grained distinctions.

In an extreme experiment, we eliminated the class name entirely, constructing prompts solely with descriptors. This approach caused a sharp decline in accuracy, highlighting the critical role of class labels in guiding the model to differentiate between categories effectively. Without class names, the model struggled to perform reliable classification, even with detailed descriptors.

Finally, we replaced descriptors with repeated class names, emphasizing the role of the class in the prompt. This modification significantly improved accuracy, showing that class labels play a vital role in the model's performance by providing clear, consistent information for classification. These findings underscore the importance of thoughtful prompt design, particularly the positioning and inclusion of class labels, in achieving optimal performance. The summary of all prompt structure modifications, along with their respective accuracy results, is presented in Table 6.

### 6.4.2 LENGTH OF PROMPT

Table 7: Impact of Length of Prompt on Zero-Shot Classification Accuracy

| Method | Accuracy (%) |
|---|---|
| CLIP Baseline | 51.95 |
| D-CLIP GPT-3 Baseline | 52.57 |
| D-CLIP GPT-4 Baseline | 53.90 |
| Random character count: 2 | 51.55 |
| Random character count: 5 | 51.87 |
| Random character count: 10 | 51.10 |
| Truncation (Class label only) | 51.78 |
| Truncation (Maximum @ 100%) | 53.88 |
| Truncation (Minimum @ 10%) | 50.77 |
| Truncation (@ 0%) | 52.23 |
| Truncation (@ 50%) | 51.34 |
| Truncation (@ 70%) | 53.59 |

In this section, we conduct experiments to examine the influence of prompt length on classification accuracy, independent of semantic content.

In the first experiment, we control prompt length by truncating descriptors to specific fractions of their character count while maintaining the overall prompt structure. For example, truncating a 100-character descriptor to 20% retains only the first 20 characters. Full descriptors correspond to 100% truncation, while 0% truncation leaves only a minimal structure with the class label and punctuation (e.g., "Black-footed Albatross,"). A "class label only" baseline prompt is also tested to isolate the descriptor's impact on accuracy.

Results show accuracy decreases with progressive truncation, with a minimum observed at 10–20% truncation. Notably, the "class label only" prompt performs worse than even minimally truncated descriptors, highlighting the value of partial descriptor information.

In the second experiment, we isolate the effect of length by appending random strings to class labels (e.g., "Black-footed Albatross ghdf idfh"). This ensures that only character count varies, enabling us to assess how prompt length, independent of semantic content, influences accuracy. In Table 7, we present a summarization of truncation levels and prompt length, along with their corresponding classification accuracies.

## 7 CONCLUSION

We propose a novel method, DefNTaxS, that enhances zero-shot image classification using VLMs by refining the taxonomic structure of classes, further enhanced by regenerating class-specific descriptors. Our method significantly improves classification accuracy across various image classification benchmarks, outperforming several state-of-the-art methods. We conduct a comprehensive evaluation of the proposed method, including domain generalization experiments, ablation studies, and comparisons with existing approaches. Our results demonstrate the effectiveness of DefNTaxS in improving zero-shot image classification performance and generalization capabilities. The proposed method provides a systematic approach to enhancing the interpretability and accuracy of VLMs for image classification tasks, offering valuable insights into the model's decision-making process.

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

# A APPENDIX

## A.1 GENERATING SUBCATEGORIES

The first step involves generating an initial list of subcategories for the dataset's classes. A context prompt is used to instruct the LLM to group the classes into subcategories, formatted as a Python list.

**Prompt:**

```
The [DATASET_NAME] dataset is constructed from [NUMBER_OF_CLASSES]
    classes. You will create at minimum [MIN_SUBCATEGORIES]
    subcategories to group the classes by and assign at maximum
    [MAX_CLASSES_PER_SUBCATEGORY] of the [DATASET_NAME] classes to each
    subcategory. For an example of a subcategory and its classes, a
    subcategory "kitchen utensil" may have the classes "fork", "knife",
    "can opener" and "teaspoon" assigned to it. Every class must be
    assigned to a subcategory, none can be missed.

First, create the list of subcategories to assign these [DATASET_NAME]
    classes to, in the exact form of a Python list and nothing more, and
    stop there before assigning the classes.

[DATASET_NAME] classes:
[CLASS_LIST]
```

## A.2 REFINING SUBCATEGORIES

If the generated subcategories are too broad or lack specificity, they are refined to ensure better granularity. The prompt requests LLM to break down broad categories into finer ones for better differentiation among classes.

**Prompt:**

```
The subcategories in this list are too coarse and will not differentiate
    the classes well. Break down the existing subcategories into more
    specific subcategories to better group the classes, e.g. instead of
    \"dog\" and \"cat\", use \"terrier\", \"retriever\", \"siamese\" and
    \"persian\". Use as many as needed to allow the classes to be as
    distinct as possible, and even removing overly broad subcategories
    like \"dogs\" and \"cats\". Once again, do not assign classes yet.

Subcategories:
[CATEGORY_LIST]
```

## A.3 ASSIGNING CLASSES TO SUBCATEGORIES

In this step, each class in the dataset is assigned to the most appropriate subcategory from the refined list. LLM is instructed to select a subcategory for each class without introducing new categories.

**Prompt:**

```
Which of the subcategories in the above Python list should
    '[CLASS_NAME]' be assigned to? It must be one of the subcategories
    in the list, not a new one. If a class could belong to multiple
    subcategories, assign it to the most unique/least likely
    subcategory. Respond with only the subcategory name.
```

