# OpenReview forum: "DefNTaxS: The Inevitable Need for More Structured Description in Zero-Shot Classification"
_ICLR.cc/2025/Conference — Submitted to ICLR 2025_

### Official Review · Reviewer_ewH2 · 2024-10-22

**Soundness:** 2
**Presentation:** 3
**Contribution:** 2
**Rating:** 5
**Confidence:** 4

**Summary:**

The paper presents a novel method to enhance prompts for zero-shot classification tasks by leveraging a large language model (LLM) to identify taxonomic relationships between classes. By incorporating the taxonomy of a given class into the prompt, the method mitigates competition between semantically similar classes, leading to more accurate classifications. The proposed approach is rigorously evaluated against established baselines across multiple classification datasets.

**Strengths:**

- The paper is clearly written and easy to follow, with a logical flow that makes the concepts accessible.
- Extensive experimental evaluations provide valuable insights into the method's performance and applicability.
- The proposed method is both simple and well-motivated, making it easy to implement and understand.
- The method demonstrates consistent improvements across the majority of evaluated datasets, showcasing the method's effectiveness.

**Weaknesses:**

- The proposed method offers limited novelty compared to existing literature. Specifically, WaffleCLIP already introduces the idea of incorporating one high-level concept into the prompts. If my understanding is correct, the baseline used for comparison seems to only include the random characters/words, without accounting for these high-level concepts. It would be interesting to see how the proposed method compares to WaffleCLIP under these conditions.
- [Minor, subjective] Figure 1 could benefit from a layout adjustment, such as adopting a more landscape-oriented aspect ratio.

**Questions:**

- CuPL is mentioned as a baseline in Section 4.2, but is never compared against. Could you explain why it was omitted from the experimental comparisons?

- The paper states in two instances that high intra-class similarity contributes to model confusion. My understanding is that high intra-class similarity means that images within the same class are visually alike, which should benefit classification. Could you clarify this point?

- In Section 5.1, it is claimed that the largest improvements occur on datasets with either high-class counts or high intra-class similarity. However, Table 1 shows that the most significant improvements are observed on Oxford Pets and EuroSAT, both of which have relatively few classes and exhibit high inter-class similarity. Did I miss something here, or could you clarify this discrepancy?

- It is claimed that appending taxonomy classes to prompts helps reduce inter-class competition and improves classification accuracy. However, since inter-class similarity typically occurs between classes within the same taxonomy, I would expect that the addition of these taxonomy-based prompts would not necessarily mitigate this issue. Could you provide more clarification on how the method addresses inter-class similarity?

- To gain insights into the above issue, a potential experiment could involve measuring the impact of DefNTaxS on the type of classification errors. Specifically, the experiment would assess how often false predictions occur within the defined taxonomy versus those occurring outside of it.

---

> ### Author Response · Authors · 2024-12-04
> **Response to ewH2**
>
> Thank you for taking the time to give feedback and adding references to clarify the paper. We’ll aim here to give some extra context to our work and respond to some of your points directly.
>
> **W1.1 - Use of WaffleCLIP Concepts**
> We have added WaffleCLIP + Concepts as a benchmark, using the high-level concepts provided in Roth et al. 2023 for the CUB, EuroSAT, Places365, Food101, and Oxford Pets datasets. While these concepts are limited to a select number of datasets and apply only at the whole-dataset level, DefNTaxS is able to create these hierarchical labels to subcategories within these datasets, using them to create distinction, increased interpretability, and ultimately improve performance. Our approach also automates the definition of these taxonomies, as opposed to the level of manual effort involved in generating these in WaffleCLIP. Please see responses to other reviewers for comparisons to other approaches (character count is very limited).
>
> **W1.2 - Formatting Issues**
> Figure 1 has been condensed to reduce space usage. Also, equations have had reference numbers added, and bolding and underlining have been added to experimental results tables to indicate performance.
>
> **Q1 - CuPL Benchmarking**
> This may have been a simple omission in the Python-to-LaTeX printing code. CuPL has now been included in Table 1 of the submission as a baseline.
>
> **Q2 - Intra-class Similarity**
> In this case, we are referring to similarities in class labels or descriptors between classes that are actually not considered similar and can be resolved through extra context, e.g., "crust" describing the Earth or a pie, "boxer" describing a dog or an athlete, etc. In the situation you are describing, we agree, and we see that maybe our terminology may have been more precisely defined in this case.
>
> **Q3 - Improvements on EuroSAT and Oxford Pets**
> We have added further context to the additional changes to the text prompts for these two specific datasets, but both can be described through improved contextualization. Results for EuroSAT showed similar improvements to any other dataset when applying the taxonomic subcategory generation equally, but once terms like "satellite imagery" or "images from EuroSAT satellite" were added, we observed the performance jump seen in Table 1. We surmise that this is due to the significant difference between the semantics of the class labels in this dataset when seen from Earth versus from orbit. Similarly, the Oxford Pets dataset received a significant boost from the inclusion of the E-CLIP templates, a factor unique to this dataset and described in lines 331-338.
>
> **Q4 - Reduction of Competition Between Classes**
> Leveraging the response to Q2, this is largely due to incorrect correlations between otherwise dissimilar classes. In other cases like fine-grained datasets such as CUB, the taxonomic subcategory labels like "small birds" and "large birds" may be generated. These act as consistent separators between classes that may otherwise share descriptors like "black bird", "orange beak", and other extremely common descriptors.
>
> **Q5 - Additional Experimentation**
> This experiment was intended to be undertaken for this submission, but we unfortunately ran out of time. We expect to see that the errors of misclassification while classifying correctly within a dataset are improved with DefNTaxS in comparison to other baselines used in this submission.

---

### Official Review · Reviewer_cME6 · 2024-10-29

**Soundness:** 2
**Presentation:** 2
**Contribution:** 1
**Rating:** 3
**Confidence:** 4

**Summary:**

The authors propose Defined Taxonomic Stratification (DefNTaxS), a novel method that leverages the power of large language models (LLMs) to enhance zero-shot image classification in vision-language models (VLMs) like CLIP. DefNTaxS introduces a hierarchical classification framework by generating subcategories that group semantically similar classes, reducing competition between classes with overlapping features. The authors demonstrate that DefNTaxS sometimes outperforms existing methods across various benchmark datasets, including ImageNet, CUB, Oxford Pets, DTD, Food101, Places365, and EuroSAT and model sizes. Furthermore, they perform some ablation studies highlighting the importance of taxonomic refinement, prompt structure, and incorporating richer descriptors for achieving performance.

**Strengths:**

- The paper is clear, well-structured, and easy to follow.
- The proposed method demonstrates improved results across all CLIP sizes on certain datasets, such as Oxford Pets and EuroSAT.

**Weaknesses:**

- The authors mention WaffleCLIP in the introduction but largely overlook its core finding: that adding additional words around the class name in prompts for CLIP has minimal effect, and previous works ([1],[2]) showing improvements may not provide meaningful benefits.

- The main results (Table 1) indicate some improvements on specific datasets and CLIP model sizes; however, these gains are generally minor and may be due to hyperparameter optimization rather than the method itself. Additionally, the proposed method underperforms compared to others on a substantial portion of the datasets.

- If I understand correctly, Table 2 shows that the method often does not outperform other approaches. For model sizes B/16 and L-14, D-CLIP achieves slightly better results, but the differences are negligible.

- Minor typos: Line 40 - "also" is split; Figure 1 - "visualisation" should be "visualization"; Line 47 - "labelsWhile".

**Questions:**

-


[1] Sachit Menon and Carl Vondrick. Visual classification via description from large language models. ICLR, 2023.
[2] Sarah Pratt, Ian Covert, Rosanne Liu, and Ali Farhadi. What does a platypus look like? generating customized prompts for zero-shot image classification.

---

> ### Author Response · Authors · 2024-12-04
> **Response to cME6**
>
> Thank you for taking the time to give feedback and adding references to clarify the paper. We’ll aim here to give some extra context to our work and respond to some of your points directly.
>
> **W1 - WaffleCLIP Findings**
> While WaffleCLIP concludes that their work is simply "a sanity check" and that "VLMs struggle to leverage the actual semantics," their experiments in (1) 4.3.2 suggest that despite similar accuracy scores, D-CLIP and their own approach tend to produce these results through very different mechanisms (with the only difference in approach being semantic information vs noise) and (2) that WaffleCLIP + Concepts tends to "demonstrate consistent and significant improvements." Other experiments of ours show that simply removing either the fine-grained semantic descriptors or the taxonomic hierarchical information from the prompt significantly alters performance in a manner consistent per dataset, suggesting both that semantics have more effect than previously suspected and that CLIP incorporates a level of hierarchical language structuring within its representation space.
>
> **W2 - Context of Improvements**
> As can be observed with the other baselines, the scale of relative incremental improvement is consistent with other literature in this field. DefNTaxS achieves equal or greater performance compared to all other baselines on the tested datasets (Table 1). All hyperparameters shared across approaches were kept consistent for fair comparison, and hyperparameters introduced in this publication (e.g., minimum number of classes per taxonomic class) are all calculated and selected as part of the taxonomy creation process (Figure 2 and lines 157-209).
>
> **W4 - Error Correction**
> These minor typos have been addressed, thank you for picking up on those and apologies that you had to. Other formatting changes include:
> - Figure 1 has been condensed to reduce space usage.
> - Equations have had reference numbers added.
> - Bolding and underlining have been added to experimental results tables to indicate performance.

---

### Official Review · Reviewer_Uvhk · 2024-11-04

**Soundness:** 3
**Presentation:** 2
**Contribution:** 2
**Rating:** 5
**Confidence:** 4

**Summary:**

This paper proposes to leverage LLMs to generate hierarchical taxonomic sub-categories for a specific category to augment the textual prompt of CLIP to better capture the semantic information in the text prompts as well as refine the alignment between the images and prompts. Experimental results on zero-shot classification validate the effectiveness of the proposed prompting method. Further, fine-grained investigations are conducted on several factors, e.g., the prompt lengths, and prompt formats.

**Strengths:**

Strengths:
- Fine-grained analysis and investigations are conducted on the different factors of building good semantic prompts, which can offer readers with some practical inspirations.
- The performance gains on some datasets are good.

**Weaknesses:**

Limitations:

- Does not provide detailed qualitative examples of the proposed prompt contexts.
- The technical novelty of this paper is limited, considering a bunch of existing works on augmenting the CLIP textual prompts. Therefore, the technical differences between this work and other works on LLM-based prompt augmentation methods need to be clarified.
- Lack of baselines for comparisons: two important baselines, CHiLS and MPVR, are mentioned in the related work but are not compared. Besides, there is also a line of missing related works on prompt augmentation with semantic discriminativeness, e.g., S3A[1], Meta-Prompting[2], and LLM Explainer [3].
- Technical issues: (1) It is not guaranteed that the LLM-generated subcategories can satisfy the completeness and disjoint constraints of taxonomy stated in section 3.1. In practice, to what extent these two constraints can be satisfied and what implications will it have on the results needs further investigations and discussions. (2) How many layers of generated taxonomy will there be? If only generate a single layer of subcategories, this work would have high technical similarity with CHiLS; otherwise, more ablations are needed to investigate its benefits.
- Overclaim issue: (1) the motivation of the hierarchical taxonomic prompt starts from the game theory, however, the competition and players in the categorization contexts are not clearly defined. There is no clear and direct relationship between them. (2) The claimed semantic interpretability advantage mentioned in the abstract is not supported since there are neither interpretability results and comparisons nor qualitative examples.
- Presentation issue: The motivation figure 1 covers too large areas. Can add some subtitles to the paragraphs in section 6.3.


References:

[1] S3A: Towards Realistic Zero-Shot Classification via Self Structural Semantic Alignment

[2] Meta-Prompting for Automating Zero-shot Visual Recognition with LLMs

[3] LLMs as Visual Explainers: Advancing Image Classification with Evolving Visual Descriptions

**Questions:**

All my questions are listed in the weaknesses.

---

> ### Author Response · Authors · 2024-12-04
> **Response to Uvhk**
>
> **W1 - Detailed Prompt Context Examples**
> Examples of g(C,t_i) have been provided in lines 235-241, including Eq. 6, and context added in Figure 2 (lines 162-179).
>
> ---
>
> **W2 - Distinction from Existing Literature**
> The distinction between DefNTaxS and the existing literature is in the method of implementation of the LLM-derived taxonomic subcategories, specifically their direct use in the classification text prompt alongside fine-grained descriptors. This approach incorporates hierarchical elements directly into the text embedding, instead of using them iteratively to reweight the model prior to classification (CHiLS, Novack et al., 2023) or creating text elements for classification separate from the class labels (ChatGPT-Powered…, Ren et al., 2023). These elements rely on non-interpretable clustering methods and modified scoring methods.
>
> DefNTaxS balances efficiency and accuracy of the LLM in creation and refinement of taxonomic subcategories, avoiding errors in generation as seen in D-CLIP descriptors (Menon et al., 2023, Roth et al., 2023). By incorporating both class-specific descriptors and taxonomic contexts into prompts, this approach increases the semantic granularity of zero-shot classification, improving interpretability, accuracy, and robustness to descriptor noise. It also gives insight into the taxonomic representations implicit in CLIP’s language structures through the effect of varying levels of visual and taxonomic granularity being more effective than either individually (Ablation 6.3).
>
> ---
>
> **W3 - Baseline Selection**
> *MPVR* is a concurrent work that has significant benefits within this field of work, but was being released alongside the completion of our work, making it difficult to recreate for comparison within the timeline of this submission. We acknowledge that there appears to be a marginal relative improvement on most datasets using MVPR, but it comes with a cost. MVPR uses a significantly increased text volume, making interpretability and comparison to known factors difficult. The intermediate prompting step also produces brand new prompts that affect the performance of the final classification step at each iteration, potentially leading to inconsistencies in performance and ability to externally verify this information.
>
> *CHiLS* cannot be compared to pure zero-shot methods like CLIP, D-CLIP, and DefNTaxS due to the iterative recalculation of the model weights using the LLM-generated hyponyms, creating an unclear baseline comparisons for a given model.
>
> *S3A* is a few-shot classification method (Zhang et al., 2023) and requires a user to have access to extra application-specific training data to enable the performance increases achieved in this approach. The DefNTaxS approach has been compared only to other zero-shot classification approaches for the fairest comparison possible.
>
> *LLM Explainer* is a method to provide "post-hoc explanations [...] for other complex predictive models" (Kroeger et al., 2024). While it does share the benefit of increased explainability with the DefNTaxS approach, the aims, and applications are distinct.
>
> ---
>
> **W4.1 - Disjoint and Completeness Constraints**
> To clarify our interpretation of this query on completeness and disjoint constraints, we are taking this question to ask how it can be certain that:
> 1. A class will be assigned to one and only one taxonomic subcategory, and
> 2. All classes will be assigned to a subcategory with none leftover.
>
> We have added Figure 2 and extra detail around this portion of the method section (lines 157-209) to clarify process flow, and Appendix A for the prompts used. In short, the LLM API is called iteratively for assigning each class, avoiding errors due to overloading the LLM with too many instructions at once.
>
> ---
>
> **W4.2 - Taxonomy Layers**
> Like CHiLS, only a single layer of taxonomy is generated. However, unlike CHiLS:
> - This taxonomy focuses on traversing the taxonomic hierarchy upward, allowing use on datasets that contain classes of refined taxonomy, not only high-level classes like "dogs" and "cats."
> - DefNTaxS does not alter the weights of the base model, minimizing compute and reducing potential errors/inconsistencies due to manual editing.
> - Utilizes both fine-grained semantics and taxonomic hierarchical information, allowing for grouping of classes into subcategories (especially useful in generalized datasets) while still maintaining distinction between those classes.
>
> ---
>
> **W5.1 - Game Theory Reference Adjustment**
> The references to game theory in this case were intended to be an illustrative reference to the competition between classes. This reference has been removed to prevent confusion.
>
> ---
>
> **W5.2 - Semantic Interpretability**
> As in related literature [D-CLIP, WaffleCLIP, CuPL], "interpretability" is to be understood as "able to be read and validated through natural language." Qualitative examples of g(C,t_i) have been added in Figure 2 and lines 235-242 and Eq. 6.

---

### Official Review · Reviewer_DoJ7 · 2024-11-04

**Soundness:** 1
**Presentation:** 1
**Contribution:** 2
**Rating:** 3
**Confidence:** 4

**Summary:**

This study concentrates on zero-shot classification utilizing CLIP. To boost performance, it aggregates the class set into various clusters, termed taxonomies, which function as superclasses. By adapting the prompts to the format “{c} which is/has {d}, {g(C,T_c)}”, the model demonstrates enhanced accuracy compared to the simplistic template “a photo of {c}”. Positive results are shown in some experiments.

**Strengths:**

The work presents a simple and straightforward method that changes the prompts to a more carefully designed prompt using taxonomies.

**Weaknesses:**

1.  Employing class hierarchy is not groundbreaking in few-shot classification. As highlighted in related work, both CHiLS and D-CLIP utilize hierarchy, along with other studies not cited here, such as [Ren. NIPS 2024].
*  Ren Z, Su Y, Liu X. ChatGPT-powered hierarchical comparisons for image classification[J]. Advances in neural information processing systems, 2024, 36.

2. This paper fails to elucidate crucial details, such as the process of clustering and g(C, T_c). In addition, in experimental evaluation, it appears to compare with lower baselines instead of fair benchmarks.

3. The presentation is poor.  Figure 1 occupies excessive space. Additionally, all equations lack equation numbers for reference.

**Questions:**

1.  The paper fails to elucidate the method for clustering classes, and what's g(C, T_c) exactly. As illustrated in 181, the superset will be refined, how is it refined? In addition, the proposed method employs numerous hyperparameters, such as |c|/10, |C| < 20. How are the hyperparameters decided?  The process appears quite handcrafted and lacks generalization.

2.  In experimental evaluation, this work achieved an accuracy of 68.03 on ImageNet using ViT B/16.  As shown in [CLIP ICML 2021],  its enhanced version with certain prompting variations can attain an even better result of 68.6. It suggests that the benefits might be obtained through additional prompting templates, such as "an image of {}", etc. More experiments are needed to determine whether the supplementary taxonomy can compensate for CLIP's prompting templates.

---

> ### Author Response · Authors · 2024-12-04
> **Response to DoJ7**
>
> Thank you for taking the time to provide feedback and references to help improve this submission. We have made modifications in line with your suggestions where appropriate. We’ll aim here to give some extra context to our work and respond to some of your points directly.
>
> **W1/W2 - Hierarchy in Relation to Existing Literature**
> The distinction between DefNTaxS and the existing literature is in the method of implementation of the LLM-derived taxonomic subcategories, specifically their direct use in the text prompt used for classification. This approach incorporates hierarchical elements directly into the text embedding, instead of using them iteratively to reweight the model prior to classification (CHiLS, Novack et al., 2023) or creating text elements for classification separate from the class labels (ChatGPT-Powered…, Ren et al., 2023). These elements are often more prone to semantic noise, rely on non-interpretable clustering methods, and use modified scoring methods. DefNTaxS balances efficiency and accuracy of the LLM in creation and refinement of taxonomic subcategories, avoiding errors in generation as seen in D-CLIP descriptors (Menon et al., 2023, Roth et al., 2023), and requires minimal inference compute due to only modifying text, no image modification or further training. By incorporating both class-specific descriptors and taxonomic contexts into prompts, this approach increases the semantic granularity of zero-shot classification, improving interpretability, accuracy, and robustness to descriptor noise. It also gives insight into the taxonomic representations implicit in CLIP’s language structures through the effect of varying levels of visual and taxonomic granularity being more effective than either individually (Ablation 6.3). DefNTaxS maintains all these benefits while also achieving improved or equal performance across the majority of relevant benchmarks.
>
> **W3.1/Q1 - Examples of g(C,t_i) and clustering**
> An example of g(C,t_i) is provided in lines 235-241, including Eq. 6, with additional context added in Figure 2 (lines 162-179). The LLM-based taxonomy creation and assignment method, distinct from k-means or other clustering methods, is further explained in Figure 2 and its caption.
>
> **W3.2 - Benchmarks**
> For fair comparison, we have used the same benchmarks as in existing literature on text-augmented zero-shot classification and similar fields [Menon et al. 2023, Pratt et al., 2023, Roth et al. 2023, Ren et al., 2023, Li et al., 2024]. CuPL (Pratt et al., 2023) and WaffleCLIP + Concepts (Roth et al., 2023) have also been added as benchmarks. If other relevant benchmarks were preferred, we would have appreciated if they had been named and referenced.
>
> **W4 - Formatting Issues**
> Figure 1 has been condensed. Equations have had reference numbers added. Boldface and underlining have been added to results tables to indicate performance.
>
> **Q1 - Hyperparameter selection**
> Selection of the hyperparameters in question (minimum taxonomy count, maximum class count per taxonomy) was conducted through empirical selection. This result generalized to all datasets tested. Reduced performance resulting from changes to this approach is shown in Ablation 6.1. These hyperparameters are also not restrictively enforced; they are simply provided to the LLM to use as a guide, as in Appendix A.
>
> **Q1 - Superset (subcategory) refinement**
> More detailed explanation of the refinement method has been provided in Figure 2 and its caption. The LLM-based taxonomy creation and assignment method is simply repeated with adjusted hyperparameters if certain outcomes are not met.
>
> **Q2.1 - Varying Results for CLIP/E-CLIP**
> It is suggested that CLIP and E-CLIP achieve different results in the original OpenAI paper where it was introduced. Each paper referenced in this submission also reports different results for these values. For example, for the ViT-B/32 backbone, CLIP's performance on ImageNet ranges from 58.5 (D-CLIP, ChatGPT-Powered) to 63.4 (CuPL). As stated in Section 4.3 (Baselines), lines 294-297, we recreated all these benchmarks using the code provided in these works to ensure a fair comparison. We achieved a result of 58.9 on ImageNet using ViT-B/32, comfortably within this range, suggesting a fair comparison of results to these other baselines.
>
> **Q2.2 - Experiments to Combine CLIP Templates with DefNTaxS Approach**
> Experiments were conducted combining elements of various baseline approaches with DefNTaxS, including D-CLIP descriptors, random characters, and dataset-level concepts from WaffleCLIP, and prompt templates from E-CLIP. None produced performance improvements or scientifically useful results, so were omitted in favor of more valuable inclusions. In specific response to this suggestion, using the E-CLIP prompting templates in combination with DefNTaxS reduced performance for all benchmarks except Oxford Pets, lines 331-338.

---

### Author Response · Authors · 2024-12-04
**General Response**

Thank you all for the time and effort putting into reviewing our submission and offering your improvements. We hope the findings of the work to be interesting and valuable to our understanding of zero-shot classification performance, but also the implications on the structure of the representation space of multimodal models like CLIP.

We look forward to your decision for this submission while wishing you all happy holidays.

---

### Meta-Review · Area_Chair_b8sh · 2024-12-15

**Metareview:**

This paper addresses the task of zero-shot classification by leveraging a large language model (LLM) to identify taxonomic relationships between classes. The review appreciated the method’s simpleness, tut they also raised important concerns including insufficient comparison, limited novelty, overclaim issue,etc. The authors did provide additional results in the response, but more experimentation is required to fully justify the framework.  Overall, the drawbacks outweigh the benefits of the paper.

**Additional Comments On Reviewer Discussion:**

During rebuttal, the concerns raised by reviewers mainly include limited novelty, unclear experimental details, poor presentation, moderate performance improvement, overclaims etc. The authors provided rebuttals relatively late, and the reviewers didn’t participate in the discussion. Only reviewer ewH2 provided his final rating. Although some results and explanation are provided in the rebuttal, the novelty of this paper is limited and the performance improvement is moderate.

---

### Decision · Program_Chairs · 2025-01-22

Reject